# From Idolatry to *Gentilidade*: Assessing Local Christians' Religious Offences in the Goa Inquisition (17th Century)

## Miguel Rodrigues Lourenço

CHAM, Faculdade de Ciências Sociais e Humanas, Universidade NOVA de Lisboa, 1069-061 Lisboa, Portugal;
mjlourenco@fcsh.unl.pt

**Abstract:** During the first half of the 17th century, the Goa Inquisition increased its focus on religious offences committed by the so-called *Cristãos da Terra* (local Christianized populations). Many of these perceived offences occurred in connection with rituals, practices and behaviours stemming from Asian cultural and religious settings, leading the inquisitors in Goa to assess a variety of external features and performances ("signs") in order to determine the seriousness of the offence and the penalty to impose. While these actions were primarily labelled as "idolatry", during the 1620s, inquisitorial personnel in Goa suddenly adopted a new designation—that of "*gentilidade*"—to refer to a type of offence that involved apostasy from Catholicism in favour of the "Law of the Gentiles." In this paper, I will analyse the context that led to this epistemic change in labelling religious offences, while also comparing the extant Goa Inquisition trials and summaries with later catalogues of cases where offences first began to be designated as "*gentilidade.*" I will argue that during the 1620s such changes in classifying religious offences were the outcome of a debate that, even though it was external to the Goa Inquisition, incidentally questioned some of its procedures and prompted its inquisitors and prosecutor to repurpose an already existing term into a broad category denoting heresy and apostasy, thus reinforcing the legitimacy of the tribunal's judicial practices.

**Keywords:** apostasy; *Gentilidade*; heresy; idolatry; legitimacy

## 1. Introduction

In 2013, the authors of the first general history of the Portuguese Inquisition identified key moments of transformation for the Tribunal, not only in terms of its internal organisation, but also in terms of reorienting or recalibrating its scope of surveillance. The way in which the Holy Office directed its attention to behaviours that were not previously listed in catalogues of heresy showed its ability to adjust to the challenges posed by a changing society. Accordingly, Giuseppe Marcocci and José Pedro Paiva identified moments when other heresies (new "enemies") became more visible within the dominant ideological framework of anti-Judaism, such as Molinism, Sigilism or, later, the freemasons (Marcocci and Paiva 2013, pp. 14–19). Although none of these acquired the centrality that Judaism did as a defining offence in inquisitorial discourse, it is clear that the Inquisition was an attentive observer of the spiritual and social concerns of its time and was not indifferent to them.

Similar phenomena may be observed in the branch of the Inquisition in Goa. Created in 1560 within a context of institutional renewal and territorial reorganisation of the Holy Office, the Goa Inquisition was the only Portuguese inquisitorial tribunal established in an entirely colonial setting (Cunha 1995; Paiva 2017). Founded to operate in the *Estado da Índia*, the political-governmental body that oversaw Portuguese interests between Sofala, in what is now Mozambique, and Japan or the Moluccas, this tribunal oversaw a territorially fragmented district with its communications regulated by the monsoon season and its boundaries coextensive with those of the dioceses founded under the auspices of Portuguese patronage (Lourenço 2015). In this administrative framework, the demand to promote and protect missionary activity was a central political goal that the Inquisition also

engaged in (Marcocci 2011a, pp. 82–85; Marcocci and Paiva 2013, pp. 111–19). In the *Estado da Índia*, the largest group within the inquisitors' jurisdiction were the *Cristãos da Terra*, literally "Christians of the Land", a general designation that comprised both neophytes and individuals born into Catholicism who were potential defendants with varying levels of knowledge of the precepts of the faith. While the Goa Inquisition was not the only tribunal to exert jurisdiction over colonial territories—the Inquisition of Lisbon's district comprised the Crown of Portugal's American and African possessions—the fact that its seat was established overseas to encompass an entirely colonial inquisitorial district meant that it was faced with a number of issues specific to the *Estado da Índia*. Missionary controversies, the participation of inquisitors in the *Estado*'s other forms of government, and vigilance over local Catholics' involvement in native rituals were all problems that were unique to the Goa Inquisition. Because the inquisitors of the Lisbon tribunal were an ocean apart from Portugal's African and American possessions, where local customs also influenced the lives of Christianized populations, issues like missions, conversion, or participation in other state institutions never generated the kind of discussions it did in Goa. For this reason, the Goa Inquisition was the scene of challenges that were particular to the context of the Portuguese Holy Office, derived from the intertwining of colonial society and the tribunal.

After four decades of inquisitorial activity, such challenges came to reflect on the institutional life of the tribunal, to the point that the inquisitors increased scrutiny over local Christianized populations and their behaviours associated with traditional rites and customs. This moment of transition in Goa, from a tribunal focused on the repression of Judaism to the surveillance of a set of crimes that the historiography has conventionally called "*gentilidades*", is well known. However, analyses of the lexicon employed by the inquisitors concerning these behaviours are still few (Tavim 2016; Marcocci 2018; Lourenço 2021; Silva 2022). Yet to be understood are the variations in the lexical apparatus of the inquisitors in Goa, which resulted in the predominance of the word "*Gentilidade*"— formed from the term *Gentio* (Gentile) and initially used to refer to the latter's social and religious milieu—over that of "idolatry" when referring to the main transgressive behaviour committed in the context of this inquisitorial district. The aim of this article is to analyse, on the one hand, the Goa Inquisition's process of defining the quality and seriousness of the offences committed by the *Cristãos da Terra* and, on the other, to assess the significance of lexical change as a device for legitimising its own activity. First, I will consider the growth of idolatry as an increasing area of concern in the discourse and praxis of the Goa Inquisition. Second, I will assess the emergence of a new defining offence in the practices of classifying crimes by this tribunal, that of "*gentilidade*". This term will not only relegate the use of the term "idolatry" to other types of documents, such as the trial proceedings, but it will ultimately overlap the latter in the correspondence sent by the inquisitors to the Inquisitor General and the General Council of the Holy Office in Lisbon. Finally, I will analyse the apparent contradiction between the growing importance of "*gentilidade*" in the history of the tribunal and the less rigorous procedure maintained by the Inquisition against *Cristãos da Terra*, the type of defendants most often condemned for this form of transgression.

## 2. Reconfigurations of the Goa Inquisition I: From Judaism to Idolatry

In 1608, the inquisitors of Goa reported to Pedro de Castilho, Inquisitor General of Portugal, on their experience in judging the *Cristãos da Terra* by commenting on the large number of trials that arose every year against these local Catholics. The inquisitors referred to the Naroá Pass (*Passo*), a crossing point between the island of Divar and the territories of "*Terra Firme*" beyond the confines of the *Estado da Índia*.[1] Here, they wrote, "almost all of them are idolaters, which is the Judaism of this place" (Baião 1930, p. 373)[2].

The inquisitors' equation between idolatry and Judaism is striking, especially if we consider the importance granted to surveilling the latter offence, not only at the founding of the Inquisition in Portugal, but as the core of its concerns during a long period of its history (Marcocci 2011b; Marcocci and Paiva 2013, pp. 49–76). In the Goa Inquisition,

Judaism was a central concern during the first two decades of its activity, as reflected in the correspondence and percentage of cases, reaching 42% in the 1570s for instance (Silva 2018, p. 84).

The Goa Inquisition shared the same concern for rooting out "concealed Judaism" as the other inquisitorial tribunals located in Portugal, to the point that the decline in new trials against this crime caused discontent among the inquisitors. For example, in 1586 Inquisitor Rui Sodrinho de Mesquita alerted the General Council of the Holy Office of the difficulty in identifying new cases of Judaism in the city. Thus, it was desirable to direct attention to more distant territories, such as Malacca or Macau, where there were reports of a strong New Christian presence (Baião 1930, p. 111). This issue might even have been the cause of some embarrassment for Mesquita, after all, he had been transferred to Goa in 1584 to succeed Bartolomeu da Fonseca, the inquisitor who had single-handedly led the most intense wave of repression of Jewish offences to that point (Lourenço 2022, p. 41). Admitting the Goa Inquisition's inability to scrutinise new cases of Judaism, a crime so central to the Inquisition's rhetoric of legitimacy, would certainly not be the report that a newly initiated inquisitor would want to send. Notoriously, former Inquisitor Fonseca's actions had succeeded in striking such a significant blow to the community of New Christians in the city that denunciations against them had dropped significantly.

The fact that, in 1608, the inquisitors wrote to Lisbon declaring that in Goa another crime had taken the place that Judaism had previously held in the kingdom, means that, in the 20 years that had passed since Rui Sodrinho de Mesquita's 1586 letter, behaviour linked to the ritual practices of local cultures had become the focus of the Goa Inquisition's attention. According to the judges, however, the severity of the offence was not the same. In the same 1608 letter, inquisitors warned that their predecessors had released defendants accused of these offences on bail due to their lack of Catholic education and poverty and despite being tried as heretics and apostates, "because there is no communion of their errors among them; nor any of the other reasons that can be considered in Judaism" (Baião 1930, pp. 370–71).[3] It is clear that, at the beginning of the 17th century, the inquisitors did not consider the transgressions of the *Cristãos da Terra*, nor the sociabilities framing these, as fitting the parameters to classify them as heresy, for example, maliciously communicating errors against the faith to others.

The shift in the tribunal's focus from Judaism to idolatry took place gradually towards the end of the 16th century. Luiza Tonon da Silva points out that "*gentilidades*" made up 20% of the cases judged by the Goa Inquisition between 1581 and 1590 and 34% between 1591 and 1600, rising to an astonishing 68% in the decade in which the 1608 letter was written (Silva 2018, pp. 77–89).[4] From the beginning of the 17th century, thanks to increased collaboration with the Archbishop of Goa Fr. Aleixo de Meneses (Beylerian 1974, pp. 591–92), and due to the Provincial Council of 1606, the atmosphere was certainly favourable to shift attention to idolatry. In fact, the very mention of the Naroá Pass reflects a decades-long debate in the Provincial Councils of Goa—including the most recent one in 1606—about the connections between *Terra Firme* and the *Estado da Índia* at festival times, an opportunity for local Christians to celebrate "Gentilic" ceremonies, but also for some individuals classified as "dogmatizers" (*dogmatistas*) or "preachers of *gentilidade*", such as *bhot, yogui* and *jyotiṣī*, to enter Portuguese domains.[5]

Given the importance that inquisitors placed on "idolatry", it is surprising that this type of crime is practically absent from the *Reportorio* (directory) drawn up in 1623 by the then prosecutor João Delgado Figueira, in which he recorded the tribunal's judicial activity between 1561 and 1623.[6] Of the cases listed by Figueira, we find only one unequivocally classified as "idolatry" and two others whose descriptions resorted to the verb "idolise", very unimpressive figures for an offence that was defined as "our Judaism" (Lourenço 2021, pp. 217–18).

However, the documentation composed by the Goa Inquisition reveals a broader practice of categorising offences for "idolatry" than the *Reportorio* might at first suggest. In fact, the inquisitors reported judging individuals for idolatry on more than one occasion,

for example João Martins or Ana Fernandes ([Baião 1930](#), pp. 547, 566). However, the categorisation used by the inquisitors in their letters does not always coincide with those Figueira recorded in the *Reportorio*. For example, in the case of Martins (1618) his entry was "*gentilidade*". As for Ana Fernandes, while her name does not appear in the *Reportorio*, the fact her offence was referred to as "idolatry" attests that this category was being used in 1619, the date of the letter mentioning her case.

In addition to these and other references in the inquisitorial correspondence about the idolatry of the converted local populations, another document provides a unique opportunity to compare the classifications used before 1623. It is a list of cases perused by the Goa Inquisition in 1609 and 1610 with their respective sentences, making it the only known list drawn up before the *Reportorio* was finalised—even if it only includes defendants spared the *auto-da-fé* ceremony.[7]

As can be seen from the table of offences below highlighting transgressive behaviour related to local ritual and devotion (Table 1), the 1609–1610 list rarely provides categories of offence. Its author—possibly the Inquisition prosecutor Baltasar do Amaral Tavares—seems to have distinguished between four forms of transgressive behaviour by the *Cristãos da Terra*: sacrifices, offerings, apostasy and idolatry. "Idolatry" was, however, the only category really employed in the list, otherwise it provided the descriptions of behaviour of each defendant, as summarised below.

**Table 1.** *Cristãos da Terra* and their descriptions or categories of offence in the 1609–1610 list.[8]

| Defendant | Category or Description of Offence in the 1609–1610 List |
|---|---|
| Bartolomeu Rangel | For saying that Lopo Rangel was an idol from the mainland, lying down at his feet and making *sumbaya* to him like an idol |
| João de Figueiredo | For digging up a treasure and sacrificing a chicken to the Devil in order to find it |
| Lázaro Fernandes | For taking two roosters and rice to sacrifice to the Devil if a treasure was found |
| Francisco Fernandes | The same |
| Bernardo Beato | The same |
| Paulo Fernandes | For having allowed a certain person to call the *pagodes* to remove the Devil that was in his body and who had gone to *Terra Firme* to ask two idols for a *fula* |
| Filipe Rodrigues | Idolatry |
| Pedro de Oliveira | Idolatry |
| Domingos Fernandes | Idolatry |
| Domingos da Cunha | Idolatry |
| Simão Furtado | Idolatry |
| Lopo Rangel | For claiming that he was himself an idol and deserved to be worshipped as such |
| Pedro de Sousa | For going to *Terra Firme* to a gentilic festival and making *sumbaya* to an idol |
| Brás de Sequeira | The same |
| André Rodrigues | The same |
| Domingos de Mesquita | The same |
| Sebastião de Meneses | For giving money and figs as an offering to an idol, for going to *Terra Firme* to make supplications to an idol and for consulting a sorcerer, making *sumbaya* to him |
| Pedro Fialho | For watching a ram being sacrificed to the Devil |

**Table 1.** *Cont.*

| Defendant | Category or Description of Offence in the 1609–1610 List |
|---|---|
| Pedro Afonso | For making a sacrifice to locate a treasure |
| Miguel Fernandes | The same |
| Sebastião Álvares | For keeping gentilic superstitions, for asking an idol for a *fula*, for offerings to an idol and for giving money for offerings |
| Paulo de Meneses | The same |
| Domingos das Póvoas | The same |
| Domingos das Póvoas | The same |
| Domingos Vaz | The same |
| Francisco Ribeiro | The same |
| Rodrigo Fernandes | The same |
| Diogo de Bragança | The same |
| Gaspar de Bragança | The same |
| Fernão Martins | The same |
| Pedro Homem | The same |
| Simão de Miranda | The same |
| Domingos Vaz, "o curto" | For having agreed with the fishermen of his village to move some idols and build them a house, for having consented to the donation of a meadow to the same idols and for having consulted them, asking for a *fula* and making *sumbaya* |
| Gaspar Fernandes | For ordering the sacrifice of a ram to the Devil in a certain plain |
| Domingos Fernandes | For taking part in the sacrifice by Gaspar Fernandes |
| António Coiro | For having witnessed the sacrifice of a ram and two roosters to the Devil |
| Gonçalo Monteiro | For joining the sect of the Gentiles, believing in the *pagodes* (idols) and entrusting himself to them |
| Gonçalo Garcia | The same |
| Agostinho Gomes | The same |
| João Mendes | The same |
| Álvaro de Barros | The same |
| João de Figueiredo | The same |
| João Maciel | The same |
| Gaspar Fernandes | The same |
| António Mazaredo | The same |
| Madalena Rangel | For kneeling down in front of her husband who was posing as an idol and making *sumbaya* to him |
| Maria Correia | For consenting to the sacrifice of a chicken to the Devil, lending money to locate a treasure |
| Maria Álvares | The same |
| Domingas Coelha | For offering figs to the Devil to locate a treasure after the Devil had appeared to him in his dreams |
| Catarina Mendes | For sacrificing two roosters to the Devil to locate a treasure |

The organising logic in the 1609–1610 list shows that its author shortened the description of the crimes for the sake of expediency, especially when their circumstances

and sentences coincided with those of the previous entry. The author repeatedly used the expression "*o mesmo*" (the same) or its Latin equivalent, "*idem*", from one record to the next. The offences were listed according to a taxonomy of Catholic religious offences, thus reflecting a judicial lexicon that refashioned local practices into the vaguest of descriptions.[9] The superficial description of the behaviour served, nevertheless, a practical purpose. As the list was meant to be reviewed by the deputies of the General Council of the Holy Office, the larger the description the easier it would be for them to appreciate the seriousness of the matter at stake. Similarly, as the deputies of the General Council were by and large unfamiliar with specific forms of the rites of the "Gentiles"—as local populations of India were predominantly termed—the descriptions proposed in the list were meant to convey unequivocal transgressions. Therefore, the fact that this list presents a description of criminal behaviour, instead of mere categories, clarifies a hierarchy of gravity between the different actions—"acts of *gentilidade*" as they were called in a letter from the inquisitors in 1606 (Baião 1930, p. 343)—which were not always understood in Lisbon. In fact, as early as 1596, the deputies of the General Council requested clarification of terms from previous lists that were foreign to the kingdom's inquisitions and made the offence unrecognisable in their eyes (Faria 2010, p. 169). This became standard practice as shown by the explanations provided by the author of the list on the meaning of words such as "*fula*" ("a rose, or a flower placed in the Idol") and "*sumbaya*" ("reverence").[10]

In the complex Catholic taxonomy of religious transgressions under inquisitorial scrutiny, sacrifices, offerings, or idolatry were behaviours existing within a system of offences against God that were subsumed in the broader understanding of *superstitio* (Campagne 2002) and which, in the discursive practice of the ecclesiastical and royal authorities in the *Estado da Índia*, tended to be referred to generically as idolatries or gentilic rites and ceremonies (Ventura 2011). However, this list shows that, when communicating with Lisbon, the inquisitors felt the need to draw up a detailed discourse on the different so-called "gentilic" rites and ceremonies, possibly as a way of clarifying what they understood to be superstitious practices, such as those involving offerings and gestures of veneration to idols in order to obtain material gain or sacrifices to the Devil, a type of offence which, depending on the form of demonic pact in question and the legal opinions on the subject, could result in an accusation of heresy (Paiva 2002, pp. 56–59), on the one hand, and, on the other hand, those which explicitly involved renouncing the faith in favour of another "belief" or "law", namely that of the "*pagodes*" (apostasy) or the veneration of the creature as God (idolatry).

The cases presented above reveal that, strictly speaking, idolatry was not the preferred choice of inquisitors for labelling transgressive behaviour, although we must bear in mind that the list does not include all people prosecuted during those two years. However, the inquisitorial correspondence suggests that this was the most common crime in those years, to the extent that the deputies of the General Council referred to it in a letter of 1601, with a lexicon and framing that was very similar to discourses on heresy:

> Your Graces will order that the *Cristãos da Terra* are not to be present at the conventicles (*conventículos*) and preaching of the Gentiles and their Brahmins, prohibiting it with the sentences and censures that you will deem fit: and you will not prosecute the Gentiles except in cases where you know that they persuade or want to persuade the said Christians into their sects and try to pervert them and have them idolise [. . .].[11]

The specific usage of "conventicles", a term that in medieval anti-heretical discourse refers to the locations where heretics met in secrecy to lead Christians astray, appears in this letter in direct correlation with the actions of "perverting" and "idolise".[12] This can be seen in the trial of the *Cristão da Terra* Francisco Rangel, conducted in 1603 by the Goa Inquisition and sent to the General Council of the Holy Office due to procedural concerns the following year (Marcocci and Paiva 2013, p. 115; Feitler 2016, pp. 104–5; Lourenço 2021, pp. 219–21). The prosecutor's accusation clearly identified the case as one of "idolatry and

apostasy".[13] Rangel was accused of sacrificing rams "in honour" of the deity Ravalnath and "another devil", as well as performing other "gentilic ceremonies" in which he called on that same deity. In the eyes of the prosecutor, this meant that the defendant would consider "the sect of the Gentiles as a good thing, and believe in the said *pagode* (idol) as they do, having it as his true God, and expecting salvation, good crops, and all the other temporal goods from it alone, and not from Christ our Lord".[14] The evidence against him aggravated the defendant's case especially considering that, in the course of the trial, he had denied the *tenção* (intention, will) to carry out the aforementioned behaviour. In jurisprudence, the defendant's intention in committing the offences would determine the difference between a conscious, voluntary and malicious departing from the true path to salvation and therefore, of a recognition of the errors of the heretics as such and an adherence to them, on the one hand, and a range of intentions that did not presuppose a renunciation of faith, and could denote a lack of knowledge or ignorance of the criminal nature of their behaviour, on the other hand.

In Goa, how the intention of the *Cristãos da Terra* should be ascertained and judged was a point of constant debate for the inquisitors, especially when judging the newly converted (in Portuguese, *novamente convertidos*).[15] From a legal point of view, the status of neophytes depended on the moment of baptism and, consequently, of Catholic instruction. Baptism in adulthood meant neophytes did not have the new faith firmly rooted in their hearts and for this reason inquisitorial praxis recommended more benign treatment (*Repertorium* 1575, p. 600). In fact, neophytes had been a matter of concern for Inquisitor General Henrique since the establishment of the Goa Inquisition. In the instructions sent to the inquisitors when the tribunal was founded in 1560, Henrique had stipulated those converts "from the sect of Muhammad or Gentiles" had to be instructed before they could be reconciled, even if they had committed crimes of heresy (Cunha 1995, pp. 299–300; Marcocci 2011a, p. 81; Marcocci and Paiva 2013, p. 110). This resonated among the inquisitors who often hesitated on the fairness of trying neophytes according to the rigour of the law, namely sentencing them to capital punishment in the event of a second lapse into heresy. By the end of the century, on more than one occasion, they were invoking the "ancient statutes" (*antigo regimento*), referring to these earliest instructions, as the most appropriate procedure to prosecute these individuals (Baião 1930, pp. 258, 271).[16] The Inquisitor General must have never felt fully comfortable with this solution, until the end of his term he sought a brief from Rome that would favour neophytes in the event they relapsed. A new procedural solution was only authorised in 1599 after Pope Clement VIII issued the brief *Sedes Apostolica*, which pardoned the application of capital punishment up to the third lapse for neophytes guilty of heresy (Marcocci 2011a, pp. 87–88; Marcocci and Paiva 2013, p. 115). However, the new procedures did not dispel all procedural doubts for the inquisitors who recognised how little education most local Christians had:

> Most of the people we try here are *Cristãos da Terra* and some are not very well educated in the faith, and all of them are very shy. If we gently interrogate them they confess to some of the acts of *gentilidade*, [but] they will deny their intention [in departing from the Catholic faith]; if we are strict and arrest them on the evidence we have against them, they will easily confess to one thing or another, and we understand that they often do it out of fear of imprisonment, rather than to unburden their conscience.[17] (Baião 1930, pp. 342–43)

This letter, dated 1606, expressed what previous generations of inquisitors had already identified: that even among the *Cristãos da Terra* who received baptism at birth, their knowledge of the Catholic faith was incipient and did not differ substantially from those who had been "baptised standing up" (*baptizados em pé*), that is, as adults (Baião 1930, pp. 4–5, 258). In Lisbon, where the deputies and Inquisitor Generals did not have direct contact with the shortcomings of Christianisation in Asia, the distinction between a *Cristão da Terra*, on the one hand, and a *novamente convertido* or neophyte, on the other, did not always receive the best attention. From the outset, the resolutions passed in Lisbon seem to

be aimed at the latter, even when they chose to use the former as a category of designation.[18] In 1608, the inquisitors had still not received an unequivocal answer from the Inquisitor General Pedro de Castilho as to the procedure to be followed with the *Cristãos da Terra* who were not neophytes. This multi-year delay led them to abandon politeness and to request a clear resolution: "please tell us whether we should practise it [the brief favouring the neophytes] indiscriminately with everyone, or only with the neophytes it refers to" (Baião 1930, p. 371).[19] It wasn't until 1610 that Castilho directly addressed the issue. He expressed his surprise at the fact that local Christians

> [. . .] have not been relaxed <u>until now</u> for relapses, nor have their goods been confiscated, even though some of them are children of Christian fathers and mothers,  and were baptised on the 8th day, and are well educated, and some  have been brought up in Colleges of the Religions [i.e., the Religious Orders] from a young age to twenty and more years: and it seems that the reasons to grant favour to those [individuals that were] baptised standing up [i.e., adults] and poorly instructed do not apply to them [those born into Christianity]: and that faults committed anew by these after the first lapse are not being reported: I would be glad if you could tell me the reasons you have for not relaxing, nor confiscating their goods, nor reporting their faults: because the brief of His Holiness clearly pertains to those who were baptised as adults.[20]

For Castilho, this practice posed a problem insofar as it meant a very loose interpretation of the 1599 Papal brief. Not only did the *Sedes Apostolica* not pertain to those that had been baptised since birth (*baptizados infantes*), it also did not preclude neophytes from being prosecuted. It only withheld a sentence of capital punishment until the third lapse. The brief was not, therefore, an obstacle to a trial in accordance with inquisitorial procedure, even against neophytes. If that was the case with neophytes, reason dictated that non-neophyte *Cristãos da Terra* should be tried in conformity with the rule of law, however paying attention to the depth of their instruction in the tenets of the faith.[21] In their reply to this letter, the Goa inquisitors stated that this was their understanding and that they would follow Castilho's instruction (Baião 1930, p. 425; Feitler 2016, p. 105). A procedure was thus defined that, in practice, equated a *Cristão da Terra* born to Catholic parents with an Old Christian of Portuguese descent, taking into consideration their education and knowledge of the faith.

### 3. Reconfigurations of the Goa Inquisition II: From Idolatry to *Gentilidade*

These were the judicial procedures of the Goa Inquisition regarding the *Cristãos da Terra* when João Delgado Figueira arrived there in 1618 to serve as the tribunal's prosecutor. Figueira, who would become inquisitor in 1624, is well known in the historiography for his role in organising the tribunal's archive and, in particular, for drawing up the aforementioned *Reportorio* of trials conducted between 1561 and 1623. It is in this hefty volume, in which the then prosecutor summarised the cases and assigned categories of offence to them, that the widespread use of "*gentilidade*", as a classification of offending behaviour, is recorded for the first time. In fact, Figueira assigned this category to no less than 745 cases (the count includes assignment either alone or as one among other charges). Similarly, where crimes of "idolatry" were mentioned in inquisitors' correspondence before, letters such as those of 1620 or 1625 were now reporting "offences (*culpas*) of *gentilidade*" (Baião 1930, pp. 579, 627). The wider usage of this term during the 1620s is all the more significant when comparing the list of 1609–1610 to the *Reportorio* of 1623 (Table 2). Doing so shows that Figueira carried out a reclassification of transgressive behaviour recorded earlier under inquisitors Jorge Ferreira and Gonçalo da Silva.

**Table 2.** *Cristãos da Terra* and respective categories of offence practised in the list of 1609–10 and in the Reportorio of 1623[22].

| Defendant | Category or Description of Offence in the 1609–1610 List | Category of Crime in the *Reportorio* |
|---|---|---|
| Bartolomeu Rangel | For saying that Lopo Rangel was an idol from the mainland, lying down at his feet and making *sumbaya* to him like an idol | *Gentilidade* |
| João de Figueiredo | For digging up a treasure and sacrificing a chicken to the Devil in order to find it | *Gentilidade* (For consulting *pagodes* to achieve their health and good success in their endeavours[23]) |
| Lázaro Fernandes | For taking two roosters and rice to sacrifice to the Devil if a treasure was found | Sacrifices |
| Francisco Fernandes | The same | Sacrifices |
| Bernardo Beato | The same | Sacrifices |
| Paulo Fernandes | For having allowed a certain person to call the *pagodes* to remove the Devil that was in his body and who had gone to *Terra Firme* to ask two idols for a *fula* | For consulting *pagodes* |
| Filipe Rodrigues | Idolatry | *Gentilidade* |
| Pedro de Oliveira | Idolatry | *Gentilidade* |
| Domingos Fernandes | Idolatry | Sacrifices |
| Domingos da Cunha | Idolatry | *Gentilidade* |
| Simão Furtado | Idolatry | *Gentilidade* |
| Lopo Rangel | For claiming that he was himself an idol and deserved to be worshipped as such | *Gentilidade* |
| Pedro de Sousa | For going to *Terra Firme* to a gentilic festival and making *sumbaya* to an idol | *Gentilidade* |
| Brás de Sequeira | The same | *Gentilidade* |
| André Rodrigues | The same | Pilgrimages |
| Domingos de Mesquita | The same | Oblations |
| Sebastião de Meneses | For giving money and figs as an offering to an idol, for going to *Terra Firme* to make supplications to an idol and for consulting a sorcerer, making *sumbaya* to him | *Gentilidade* |

**Table 2.** *Cont.*

| Defendant | Category or Description of Offence in the 1609–1610 List | Category of Crime in the *Reportorio* |
|---|---|---|
| Pedro Fialho | For watching a ram being sacrificed to the Devil | Sacrifices |
| Pedro Afonso | For making a sacrifice to locate a treasure | Sacrifices |
| Miguel Fernandes | The same | Sacrifices |
| Sebastião Álvares | For keeping gentilic superstitions, for asking an idol for a *fula*, for offerings to an idol and for giving money for offerings | Sacrifices |
| Paulo de Meneses | The same | Sacrifices |
| Domingos das Póvoas | The same | Oblations |
| Domingos das Póvoas | The same | *Gentilidade* |
| Domingos Vaz | The same | *Gentilidade* |
| Francisco Ribeiro | The same | *Gentilidade* |
| Rodrigo Fernandes | The same | Sacrifices |
| Diogo de Bragança | The same | Oblations |
| Gaspar de Bragança | The same | *Gentilidade* |
| Fernão Martins | The same | *Gentilidade* |
| Pedro Homem | The same | Offerings |
| Simão de Miranda | The same | *Gentilidade* |
| Domingos Vaz, "o curto" | For having agreed with the fishermen of his village to move some idols and build them a house, for having consented to the donation of a meadow to the same idols and for having consulted them, asking for a *fula* and making *sumbaya* | *Gentilidade* |
| Gaspar Fernandes | For ordering the sacrifice of a ram to the Devil in a certain plain | *Gentilidade* |
| Domingos Fernandes | For taking part in the sacrifice by Gaspar Fernandes | Sacrifices |
| António Coiro | For having witnessed the sacrifice of a ram and two roosters to the Devil | Sacrifices |
| Gonçalo Monteiro | For joining the sect of the Gentiles, believing in the *pagodes* (idols) and entrusting himself to them | *Gentilidade* |

**Table 2.** *Cont.*

| Defendant | Category or Description of Offence in the 1609–1610 List | Category of Crime in the *Reportorio* |
|---|---|---|
| Gonçalo Garcia | The same | *Gentilidade* |
| Agostinho Gomes | The same | *Gentilidade* |
| João Mendes | The same | *Gentilidade* |
| Álvaro de Barros | The same | *Gentilidade* |
| João de Figueiredo | The same | *Gentilidade* |
| João Maciel | The same | *Gentilidade* |
| Gaspar Fernandes | The same | *Gentilidade* |
| António Mazaredo | The same | *Gentilidade* |
| Madalena Rangel | For kneeling down in front of her husband who was posing as an idol and making *sumbaya* to him | *Gentilidade* |
| Maria Correia | For consenting to the sacrifice of a chicken to the Devil, lending money to locate a treasure | Sacrifices |
| Maria Álvares | The same | *Gentilidade* |
| Domingas Coelha | For offering figs to the Devil to locate a treasure after the Devil had appeared to him in his dreams | Sacrifices |
| Catarina Mendes | For sacrificing two roosters to the Devil to locate a treasure | Sacrifices |

A comparison between the descriptions in the 1609–1610 list and the categories of crime recorded in João Delgado Figueira's *Reportorio* reveals differences in the classification options practiced by the authors of the two documents. It is important to note that, although the two documents are little more than ten years apart, the body of judges was not the same: the inquisitors Jorge Ferreira and Gonçalo da Silva left office in 1612 and 1614, respectively, and the prosecutor Baltasar do Amaral Tavares died in the latter year. His position was held on an interim basis by the notary António Dias Gago until the arrival of Figueira in 1618. Therefore, those responsible for the practices of judging, describing and categorising the behaviours on the 1609–1610 list were no longer in the service of the Holy Office of Goa when Figueira took on the role of prosecutor and began preparing the *Reportorio*.

The dissonance between the two documents shows that, in the span of a decade, there was a change in how the Goa tribunal classified criminal behaviour associated with local rites and understood its seriousness as an offence against God. This change, as we can see, was not linear. We might have expected that, even though Figueira chose to assign new categories, he would have used the same category of offence whenever the description was the same in the 1609–1610 list. However, this was not always the case.

Figueira consistently applied the category of "*gentilidade*" to all the offences that the author of the list described as an action of passing into the sect of the Gentiles, believing in *pagodes* (idols) and commending oneself to them.[24] This implies an understanding

of the category as an act of apostasy that implied renouncing Catholicism in favour of another "belief" or "law", that of the *pagodes* or idols.[25] However, other descriptions marked "the same" on the list received alternate categories in the *Reportorio*, such as "*gentilidade*", "pilgrimages" and "oblations". These were the cases of Pedro de Sousa, Brás de Sequeira, André Rodrigues and Domingos de Mesquita, all sentenced for going to *Terra Firme* to a "gentilic festival" and making *sumbaya* to an idol.[26] Similarly, one of the descriptions on the list, which applied to 12 individuals and included keeping gentilic superstitions, asking an idol for *fula*, making offerings to an idol and giving money for offerings, merited four different categories from Figueira: "*gentilidade*", "oblations", "offerings" and "sacrifices".

The abbreviated nature of the 1609–1610 list's descriptions makes it impossible to discern the reason for the difference in Figueira's categorisation practice. Only the inquisitorial proceedings—which have not survived—could explain it. However, the legal practice of the Inquisition, which required an interrogation session called "*crença*" (belief), in which individuals were examined for their "*tenção*" (will, intention), would help to explain the changes in categorisation. In terms of behaviour, certain gestures, rites and ceremonies were considered indicators of a declarative adherence to a disapproved law (often termed "sect"), while others expressed only the desire to obtain temporal benefits. As can be seen in a query sent by the judges to the Inquisitor General, they distinguished between the "ceremonies, and rites which by law are declarative [*protestativos*] of the sect [of the Gentiles], such as *sumbayas*—which constitute the greatest act of worship of the *pagodes* (idols) there is among the Gentiles—, sacrifices of fire, blood, or any other, for which they intend to give them honour and veneration as unto God", on the one hand, and "many other superstitions, and rites, which do not induce such effective assumption, such as going to the houses of the *pagodes*, without making them *sumbayas*, attending their feasts, and dances, and other [things] of this quality", on the other hand.[27] Or, as the inquisitors wrote in 1649 or 1650, it was a matter of distinguishing between those ceremonies that did not involve "sacrifice, or offering, or idolatry, or consultation and invocation of the Devil" and superstitions that "are not heretical or declarative of the gentilic sect, and that contain no suspicion of heresy or of pact with the Devil, but only the usual superstitions of these *Cristãos da Terra*".[28] The diversity of Figueira's later classifications compared to the categories from the 1609–1610 list may thus be the result of his understanding of the differing degrees of the defendant's adherence to belief in the idols or to their expectations concerning "gentilic" rituals in similar circumstances. There is therefore a hierarchy of offences in the *Reportorio*, of which "*gentilidade*" is the epitome of apostasy and heresy. Thus, Figueira did not conduct an operation of equivalence between idolatry and *gentilidade*, but rather conceptualized a new category of religious offence to convey a specific understanding of the behaviours he intended to designate under that label. This change occurred at a discursive level, even if, in practice, the tendency remained to use the term as a substitute for "idolatry".

The depth of this change is better understood by considering the debate in which João Delgado Figueira took part shortly after his arrival in Goa, namely the theological dispute over the validity of Jesuit priest Roberto de Nobili's evangelisation methods in Madurai. In fact, his participation in the debate may actually provide the clues as for why such a change occurred. Centred around questions of whether four of the rites practised by the upper castes of the mission — the use of the "*linha*" or *yajñopavīta* on the shoulder, the *kuḍumi* (tuft of hair), the practice of purification ablutions and the sandalwood signs (*tilakas*) on the forehead — were superstitious or of political value, the dispute divided the Jesuit missionaries themselves, with the Archbishop of Goa, Fr. Cristóvão de Sá, and the inquisitors taking a stand against Nobili (Županov 2001; Aranha 2010, pp. 632–35; Aranha 2012, pp. 242–43). Pope Paul V ordered a conference to be held in Goa to address the matter, and in 1619 Figueira was called upon to put his instruction as a canonist at the service of the party opposing Nobili's methods. That year, he authored two extensive legal opinions that he presented to the conference, in which he expressed his disagreement with the interpretation that the "insignia" of the Brahmins were fundamentally political ("civil") in nature.[29] On the contrary, he argued that he found those actions to express

"veneration and declaration [*protestação*] of the Religion of the said *Pagodes* and false Gods", and were therefore "declarative signs of *Gentilidade*" (Ventura 2011, vol. 2, pp. 73, 77), placing them as indicators of heretical and apostate behaviour, since they presupposed adherence to another "law".[30] In fact, Figueira would lend his authority as the tribunal's prosecutor to his vote on the matter in the 1619 conference when he stated that the Goa Inquisition prosecuted individuals for such signs as evidence of professing *Gentilidade* and, consequently, condemned them "as heretics and [have] their estates confiscated, and some may even have been burned" (Ventura 2011, vol. 2, pp. 80–81)[31]. The 1619 conference thus allowed Figueira to refashion the transgressive behaviour of local Christians according to the same principles applied to the main heresies and forms of apostasy prosecuted by the Inquisition in Portugal, namely to Judaism and Islam.[32]

After the Inquisitor General received the opinions sent from Goa and a report stating that there was a consensus that the signs were declarative of the "Gentilic sect" (Baião 1930, p. 567), and acting on the instructions of the Pope and the monarch, he held a meeting in Lisbon that ultimately decided on validating the opposite opinion. As he reported to the inquisitors of Goa in 1621, the meeting had agreed that "the aforementioned insignia were not declarative signs of any sect, but only political, to declare and distinguish nobility and wisdom".[33] Two years later, Pope Gregory XV issued the brief *Romanae Sedis antistes* of 31 January 1623, declaring that the rites and signs were only indicative of nobility, following Nobili's position.

The pontiff's final decision caused obvious unease among the inquisitors, of whom Figueira would soon be one. The papal verdict on the Madurai signs and rites reached Goa in the wake of a controversy between the Inquisition and the governor of the *Estado da Índia* over granting authorisation to Gentiles to perform their marriage rites in the Crown's domains, something the inquisitors had been trying to prevent. In this context, to deny the "*linha*" and other "gentilic" signs and rites their declarative nature/character of adherence to a disapproved religion meant allowing a new space for the *Cristãos da Terra* to be exposed to the tangible expressions of "*gentilidade*". They took this even more seriously considering that some of these "signs" had even been the target of several prohibitions by almost all the provincial councils, the last of which had taken place in 1606 (Rivara 1862, pp. 210–11). Unsurprisingly, the inquisitors hastened to point out that the brief conveyed no dispositions concerning areas outside the Madurai mission and that it would be imprudent to apply it to the whole *Estado da Índia* (Baião 1930, p. 621).

Marcocci argued that the category of "*gentilidade*", which he translated as heathenism, "was an invention to describe the public apostasy of often-uncatechized natives" (Marcocci 2018, pp. 150–51). In fact, a lexical analysis of the documentation produced by the Goa Inquisition during the 1620s allows us not only to support this assertion, but also to determine that this invention took place during the debate on the Madurai mission and was prompted by it. In fact, the events at the beginning of the decade suggest that the ministers of the tribunal, with Figueira at their head, sought to standardise the concept of "*gentilidade*" and turn it into a common designation for classifying crimes of heresy and apostasy committed by local Christianised populations, overlapping that of idolatry as the "defining offence" prosecuted by the Goa Inquisition. The reference to crimes of "*gentilidade*" in inquisitorial correspondence dates back to these years, i.e., the 1620s, and remained in force among the tribunal's ministers in the following decade. During the inspection visit to the tribunal by Inquisitor António de Vasconcelos in 1632 and 1633, the officers of the Goa Inquisition used this term to refer to crimes of *Cristãos da Terra*.[34] With the greater use of the category, "idolatry" lost its status as the defining offence in the framework of the transgressive behaviour of the local populations, although, as suggested above, it was not completely abandoned. Father António de Andrade, the Jesuit secretary of the visit, also used the term "idolatry" as a category of crime, which suggests that, in the missionaries' perception, there was no significant difference between the two offences.[35]

#### 4. *Gentilidade*, a Heresy of no Significance?

The data presented above point to "*gentilidade*" assuming the status of a defining offence, a specific concept for specific transgressions that included consideration of intent. This makes it all the more surprising, then, that a letter from Inquisitor General Francisco de Castro in 1634 to the Goa Inquisition unequivocally stated that this tribunal did not accuse the "*Cristãos da Terra*" based on their intention (*tenção*) (Marcocci 2018, p. 145).[36] Castro's statement is perplexing because the very recent ideological construction of this offence does not seem to correspond to the judicial practice concerning that very same crime as outlined in his writing.

This letter dates from a period when sources are particularly sparse, especially correspondence sent by the Goa inquisitors. Specifically, we lack the letter of 1633, which contained the question to which Castro was replying. However, other documents provide the necessary context to this issue. As a matter of fact, the Inquisitor General's answers in the 1634 letter and their formulation include textual affinities with criticisms expressed by Inquisitor Machado during the inspection visit conducted by António de Vasconcelos in November 1632. On that occasion, Machado expressed his dissatisfaction with one of João Delgado Figueira's procedures, namely not prosecuting bigamists and *Cristãos da Terra* according to the "*tenção*", which was against the statutes[37]. The statements recorded during Vasconcelo's visit provide concrete examples of Figueira's failure to prosecute an accusation based on the "*tenção*", either due to negligence or for personal interests.[38] In fact, this resulted in several accusations against Figueira for non-compliance with the regulations of the Inquisition's statutes, a contrasting image to that provided by inquisitors' descriptions of him as prosecutor, when they regularly praised his meticulous and diligent character (Tavares 2009). However, after the death of Inquisitor Francisco Borges de Sousa in 1629, Figueira alone remained at the head of the Goa Inquisition for a year, during which time, according to the accusations of some of the tribunal's officials, he took the opportunity to consolidate his authority. The sharp contrast between the profiles of prosecutor Figueira and Inquisitor Figueira suggests that his life in Goa was not a simple binary between hardworking official and corrupt inquisitor. Rather, such a conundrum requires the more balanced approach proposed by Julio Caro Baroja and Kimberly Lynn when addressing the career and profile of the Spanish inquisitors and their "combination of careerism and ideological commitment" (Caro Baroja 1997, pp. 18–28; Lynn 2013, p. 3). A more nuanced interpretation points to the possibility that we are not so much in the presence of a man of zeal, as Bartolomeu da Fonseca undeniably was, but a person driven to advance his career, whose position in the Goa Inquisition could be explained by his initial desire to acquire a high profile to facilitate a rapid rise in the inquisitorial *cursus honorum* and, in the end, by a pragmatic management of his responsibilities in the tribunal, guided by decisions that would have taken his networks of solidarity in the *Estado da Índia* into consideration. With only two years of service as inquisitor of Goa in 1632, António de Faria Machado would have been shocked by Figueira's many deviations from procedure, which is why he was likely the one, in 1633, to raise the question to the Inquisitor General on the specific point he had spoken against during the visit: whether the *Cristãos da Terra* and the bigamists should be accused of the "*tenção*" in Goa, or not.

The practice described by Machado was considered by Inquisitor General Castro as the "style (practice) that has until now been kept by that Inquisition".[39] However, it is possible that his assessment on this particular point did not take into account the complex and nonlinear evolution of how the Goa Inquisition tried the "*Cristãos da Terra*". In fact, the changing strategies of the Inquisitor Generals concerning neophytes and the turnover of the body of ministers and officers in the tribunal and the General Council of the Holy Office, must have made it difficult to maintain an institutional memory that would allow all regulations issued by the inquisitorial leadership to be adequately maintained. For example, by the end of the 16th century, the deputies of the General Council had lost sight of the recommendations of Inquisitor General Henrique regarding neophytes, and one of his successors, António de Matos de Noronha, was forced to ask the Goa inquisitors, in

1598, for a copy of what they called the "ancient statutes", even though the minute of the document was available in Lisbon.[40] It turns out, the Goa inquisitors themselves had not kept these, and were forced to copy a version that was in the convent of St Francis.[41]

Since the establishment of the Goa Inquisition, Inquisitor General Henrique had harboured expectations of implementing a judicial solution favourable to neophytes, similar to that benefiting the *moriscos* subjected to the Spanish Inquisition, but left office without having succeeded.[42] While his successors endeavoured to obtain a brief that would allow a large number of poorly educated potential defendants to avoid the judicial consequences of relapse, the difficulties faced by the inquisitors in Goa led the Inquisitor General to issue regulations that would later generate hesitations in procedure and interpretation.

In 1584, in the midst of the debate concerning apostates living in infidel lands, Inquisitor General Jorge de Almeida sent instructions to the inquisitors on how to prosecute the different categories of defendants according to the circumstances of their presentation before the tribunal. As for the neophytes, he ordered that the same procedure be followed as applied to Portuguese or *mestiço* offspring of Christian parents: a trial in which the defendants should be asked about their "*tenção*" in order to determine the mitigating factors that would either allow for a reconciliation or not. The neophytes, however, benefited from the possibility of additional extenuating circumstances, such as their lack of education, family pressures and natural inclination to idolatry, to obtain lighter sentences. However, the 1584 regulation did not specify what such sentences should consist of.[43]

A few years later, however, the Goa inquisitors recognised the contradiction between this rule and the original instructions or "old statutes" according to which neophytes were to be educated in the faith before being reconciled, an option that the judges clearly preferred (Baião 1930, pp. 258–59). This instruction hints at concerns arising from the realisation that the regulations issued from Lisbon and the general procedures did not allow them to provide a satisfactory response to the realities they faced in their judicial activities. In fact, in 1597 the inquisitors expressed their concerns over sentencing relapsed *Cristãos da Terra* that had been converts to Catholicism for more than 20 years, and thus could no longer be considered neophytes, to capital punishment. Many, they felt, still lacked the necessary instruction to understand Catholic doctrine and should, therefore, continue to be considered neophytes before the Inquisition (Baião 1930, p. 258).

The situation became more confusing, if not from a legal point of view, at least from the perspective of the inquisitors, when the brief *Sedes Apostolica* was finally obtained, aimed solely at the juridical category of neophyte. Doubts about procedure soon arose as the inquisitors identified cases in which the strict application of the law made a conviction for relapsing lawful, but morally questionable if consideration was to be paid to the general lack of education among the majority of *Cristãos da Terra*. As we have seen, the resolution of the Inquisitor General in 1610, in practice, equated the *Cristãos da Terra* born to Christian fathers and mothers with Old Christians for the purposes of judgement, their Catholic instruction being a mitigating factor. While, for the newly converted, the renewal of the "brief of the neophytes" created an exceptional legal situation, the remaining defendants from the Christianised local populations were subject to an ordinary procedure, based on "*tenção*". The defeat of the faction opposed to Roberto de Nobili's missionary project does not seem to have altered the procedure of the Goa Inquisition in this regard, although it did raise a delicate issue as to the value of "gentilic" rites or signs as indicators of a renunciation of the Catholic faith, insofar as the signs and rites mentioned in the brief had already been explicitly referred to as expressions of apostasy in previous sentences of the Goa Inquisition, as Figueira himself stated in his vote of 1619 (Ventura 2011, vol. 2, p. 77).

The question considered by Francisco de Castro, in 1634, therefore suggests less a change in procedure in the ten years between the promulgation of the brief *Romanae Sedis antistes* and the visit to the Goa Inquisition than a change in the judicial behaviour of João Delgado Figueira, whether due to convenience, bribery or because, after more than a decade of service at the tribunal, he had come to consider the lesser relevance of this procedure considering the lack of Catholic education of so many individuals. To alter the procedure

against the *Cristãos da Terra* based on the instructions of the brief was incomprehensible in light of the Goa Inquisition's resistance to conforming to its dispositions. It would also mean going beyond them, extending the Papacy's understanding of the four rites and signs of the Madurai Brahmins, as non-declarative of adherence to the "Gentilic sect", to the diverse range of ritual practices of local Christians in the *Estado da Índia*, thus dismissing the need to accuse these defendants on the grounds of their intention/*tenção*. Unlike Castro, who deferred to the knowledge of the Goa Inquisition as to the need for an accusation by "*tenção*", Machado had no doubts when he affirmed the convenience of this procedure.[44] In view of this, the General Council declared in 1636 that "the Indians who perform Gentilic rites and ceremonies, when these are declarative and violently indicative of their sect, should be charged for their intention, because experience has shown that they usually perform them with great belief in the idols in whose cult they are performed: but they will not be charged when the rites they perform are not of this quality and nature".[45] The decision was a formal return to the status quo prior to Figueira's presidency of the Goa Inquisition, leaving behind the practice which had so upset his colleague, António de Faria Machado.[46]

## 5. Concluding Remarks

In the study of the lexical changes practised by the Goa Inquisition in the first half of the 17th century, the emergence of the category of "*gentilidade*" is a key moment for understanding the practices of labelling offences and also for the tribunal's self-representation. The comparison between the 1609–1610 list and João Delgado Figueira's *Reportorio* shows that, as a classificatory label, "*gentilidade*" was not a mere substitute for "idolatry". Figueira's lexical and epistemological operations in his *Reportorio* show that there was no direct equivalence between the two types of offence. On the contrary, it could be said that the then prosecutor sought to bring together all the behaviours "declarative of the sect of the Gentiles" under the term "*gentilidade*". At a time when some of these rites were being devalued, the gift of the voluminous *Reportorio* to the Inquisitor General, Fernão Martins Mascarenhas, illustrated the heretical and apostate nature of the many transgressions that, in previous decades, the Goa inquisitors had seen in a less serious light. In the context of the Madurai controversy, Figueira's reclassification made it possible to differentiate "*gentilidade*", religious transgressions to the divine majesty (heresy, apostasy), from other "*gentilidades*", superstitious behaviour. At the end of the day, it was a reminder to the inquisitorial authorities in Lisbon not to minimise, as they were doing at the time, the seriousness of the rites and signs that the tribunal considered declarative. Long gone were the years when the Goa inquisitors had argued that the parameters according to which Judaism was judged had no parallel in the offences of the *Cristãos da Terra*.

The decades between 1590 and 1630 were a key moment in the history of the Goa Inquisition. The increase in judgements against *Cristãos da Terra* forced the Holy Office to formulate clearer stipulations as to how to try this type of defendant, and the year 1636 was decisive in clarifying the inquisitorial procedure. But at the same time, these decades were a period of vitality in which the tribunal was forced to manage two important moments of reconfiguration of its own legitimacy. The first one, following the abrupt drop in cases of Judaism in the 1580s, leading to a shift towards crimes associated with the *Cristãos da Terra* and raising the profile of the crime of "idolatry". The second, in the context of a theological debate that would have unintended consequences for the tribunal's procedure, by stripping various "declarative signs" of the quality that suggested withdrawal from the Catholic faith. In this new moment, the emergence of the term "*gentilidade*" as a category of religious offence profoundly transformed the way the Goa Inquisition enunciated the transgressive behaviours it judged. Although "idolatry" and its verbal iterations never completely disappeared from the discursive practices of the tribunal, and even less from the ecclesiastical and administrative apparatus of the *Estado da Índia*, "*gentilidade*" as a category of offence came to have a long life in the documentation produced by the Goa Inquisition.

**Funding:** This research received no external funding.

**Institutional Review Board Statement:** Not applicable.

**Informed Consent Statement:** Not applicable.

**Data Availability Statement:** Data are contained within the article.

**Acknowledgments:** I would like to thank Jessica J. Fowler for revising the English version of this text.

**Conflicts of Interest:** The author declare no conflict of interest.

## Notes

1. Until the 18th century, the Portuguese territorial centre known as the "Old Conquests" consisted of the island of Tiswadi, where the city of Goa was located, a number of other smaller islands separated from the Indian subcontinent and the regional powers of the Deccan by small river courses, as well as the provinces of Salcete and Bardez.

2. In Portuguese: "quasi todos são jdolatras que he o judaismo de qua".

3. In Portuguese: "por não auer comonicassão de seus erros entre elles; nem nenhũa outra rezão das que se podem considerar no judaismo".

4. It should be noted that the label "*gentilidades*" was used by the author herself in line with the historiographical practice of designating behaviours linked to local rituals as such, however, this does not correspond to the classification practices actually used by inquisitors and tribunal's officials. The author points out that she grouped under this category "divinations, worshipping the devil, worshipping *pagodes* (idols), witchcraft, consulting sorcerers, consulting *pagodes*, sorcery, *gentilidade*, idolatry, idolising the devil with sorceresses, invoking the devil, oblations, pact with the devil, sacrifice, sacrifice to the devil, sacrifice to *pagodes*, making oneself a gentile, superstitions, treasure or visionary" (Silva 2018, p. 74). It should also be noted that in Portuguese texts the *pagode* has the double-meaning of pagoda and idol, but predominantly means the latter in contexts when the former are no longer extant, such as in Portuguese-ruled territories. See (Dalgado 1921 s. v. Pagode).

5. See Fourth Provincial Council (1592): decrees 6 and 7, action 2 (Rivara 1862, pp. 188–89) and Fifth Provincial Council Provincial (1606): decree 10, action 2 (Rivara 1862, p. 210). Joaquim Heliodoro da Cunha Rivara published the complete acts of the sessions of all Provincial Councils in his volume 5 of *Archivo Portuguez-Oriental*, cited here. A meaningful analysis of these episcopal assemblies is still needed. For a summary of the decrees of the Provincial Councils regarding the control of interactions with the socioreligious world of "*gentilidade*", read Ventura (2011, pp. 137–43).

6. Studies centred on this important source have been growing in the last decade, largely due to the digitised processing of its information by a team coordinated by Bruno Feitler (http://www.i-m.mx/reportorio/reportorio/base.html accessed on 20 October 2023). The *Reportorio* represents a unique source for the study of the Goa Inquisition in this period, especially considering the irreparable loss of its archive in the 19th century. Figueira summarised what he considered was the most pertinent information from the trials still extant in the tribunal's archive at the time, providing basic details of the defendants and the religious offences they committed (Tavim 1997; Feitler 2012; Silva 2018). The *Reportorio* is now kept at the Biblioteca Nacional de Portugal (BNP), call number Cód. 203.

7. The preparation of such lists was part of the prosecutor's responsibilities. In time, the Goa Inquisition produced three types of lists of defendants tried in the course of the year: those who were chosen to take part in the *auto-da-fé*; those who presented themselves spontaneously to declare their transgressions to the Inquisition; the rest of the defendants. The 1609–1610 list must have been made at a time when the prosecutors were only preparing two lists—of those who did and those who did not participate in the *auto-da-fé*—since it also includes a number of individuals who spontaneously declared their offences to the inquisitors.

8. Arquivo Nacional/Torre do Tombo (ANTT), *Tribunal do Santo Ofício* (*TSO*), *Conselho Geral do Santo Ofício* (*CGSO*), liv. 369, fols. 20–39.

9. This is not to say that inquisitors at Goa were ill-informed on the specificity and functions of local rituals. Seeing as missionaries— who often served as deputies or inquisitors of the Goa Inquisition—not only wrote treatises on what they termed the "sects" of the Gentiles but were also learned on some of the great works of Hinduism such as the *Bhagavadgītā*, it is doubtful that the tribunal did not possess the necessary resources—human or documentary—to address issues pertaining to local rituals and devotions. In fact, when the Goa Inquisition was first abolished in 1774, its archive had no less than 7 manuscript books written in "Gentilic language" (*lingoa gentilica*) in the section of censored material, which points to some level of interaction with and knowledge of literary works written in local languages. Also, while the nature and intent of this list makes it difficult to ascertain which cults or practices were performed by the defendants, extant trials provide lengthier accounts of such rituals. A broad attempt at matching inquisitorial discourse and local rituals has never been pursued and falls outside the scope of this study. General letter of Brother Luís Fróis, SJ, to the Jesuits in Portugal, 8 December 1560, from Goa (Wicki 1956, p. 803); ANTT, *TSO*, *CGSO*, liv. 462, fols. 254–254v.

10. ANTT, *TSO*, *CGSO*, liv. 369, fol. 31. In Portuguese: "hũa rosa, ou bonina que Esta pegada no Jdolo"; "reuerença". The Portuguese usually employed the term "sumbaya" (from the Malay *sembahyang*) to refer to local ritual performances of reverential greetings or honor to both individuals and deities. As for the act of receiving a flower, it likely refers to the gifts first offered to a deity in a

*pūjā* ceremony and then blessed by it (*prasāda*) and redistributed to the attendance (Wilkinson 1901, s. v. "sembahyang", 405; Dalgado 1921, s. v. "Sumbaia, zumbaia"; Dalgado 1919, s. v. "fula"; Lidova 2020).

11 ANTT, *TSO, CGSO*, liv. 100, fol. 105v. In Portuguese: "Daram vossas merces ordem pera que os christãos da terra se nam achem presentes aos conuenticulos e pregacões dos gentios e dos seus Bragmanes prohibindolho com as penas e censuras que lhes parecer: E contra os gentios nam procederam senão em caso que lhes conste que persuadem o querem persuadir a suas sectas aos ditos christãos e tratam de os peruerter e fazer jdolatrar".

12 The underlying notion of *conventicula* as gatherings of secrecy and dissent led to it being applied to assemblies of a similar nature. The same word was used to refer to the witches' Sabbath, as well as to the gathering of *Alumbrados* and Jews (Paiva 2002, p. 154; Fowler 2017, p. 6; Soyer 2019, pp. 131–32).

13 ANTT, *TSO, Inquisição de Lisboa*, proc. n.º 8916, fol. 15.

14 ANTT, *TSO, Inquisição de Lisboa*, proc. n.º 8916, fol. 15v. In Portuguese: "por boa a çejta dos gentios, E crer no dito Pagode como elles crem, tendoo por deos uerdadeiro, E esperando soo delle a saluação, boa nouidade, E todos os mais bens temporaes, E não a christo nosso Senhor".

15 Read Xavier (2008, 2011) on this category.

16 ANTT, *TSO, CGSO*, livro 100, fol. 100. In Portuguese: "As mais das pessoas com que corremos de presente são christãos da terra e alguns pouco instructos na fe, e todos muito timidos, se proçedemos brandamente com elles confessão alguns dos actos de gentilidade, e negão a tenção; se com rigor que he prendellos pella proua que contra si tem confessão facilmente logo hũa e outra cousa e entendemos que muitas vezes o fazem maes por medo da prizão que descargo de suas consciencias".

17 In Portuguese: "As mais das pessoas com que corremos de presente são christãos da terra e alguns pouco instructos na fe, e todos muito timidos, se proçedemos brandamente com elles confessão alguns dos actos de gentilidade, e negão a tenção; se com rigor que he prendellos pella proua que contra si tem confessão facilmente logo hũa e outra cousa e entendemos que muitas vezes o fazem maes por medo da prizão que descargo de suas consciencias".

18 This was the case with the response of the president of the Holy Office, António de Matos de Noronha, in 1596, when he informed the inquisitors of Goa that he had learnt that the bishop of China absolved the "*Cristãos da Terra*" in the *forum externum*. However, the commission that had been addressed to the prelate on the matter concerned the "*novamente convertidos*", i.e., neophytes alone (Lourenço 2014). Cf. Biblioteca Nacional do Rio de Janeiro (BNRJ), Inquisition of Goa, 25,1,001 nº186.

19 In Portuguese: "nos faça v. s. merce mandar dizer se o auemos de praticar indistintamente com todos, ou somente com os neophetos de que trata".

20 BNRJ, *Inquisição de Goa*, 25,1,002 nº068, fols. 142–142v. In Portuguese: "ategora nam [fora]m Relaxados, por relapsos, que fossem nem se lhe confiscam os beens sendo alguns filhos de Pays e Mães Christãos, e baptizados de outo dias, e bem instructos e alguns Criados em Collegios de Religiosos desde pouca jdade ate Vinte e mais annos em que pareçe que nam concorrem as razões de fauor qu[e se] fazem aos baptizados em pee e mal in[s]tructos: e que se nam reportam as culpas que de nouo acreçem a estes depois do primeiro lapso: folgara que me auisaram das razões que tem pera se nam relaxarem, nem confiscarem seus beens; nem lhe reportarem as culpas: porque o breue de s. santidade se entende claramente nos que foram baptizados sendo adultos". The underlining is in the original document.

21 BNRJ, *Inquisição de Goa*, 25,1,002 nº068, fol. 142v.

22 ANTT, *TSO, CGSO*, liv. 369, fls. 20–39; BNP, Cód. 203.

23 I also included Figueira's description of João de Figueiredo's case due to the fact that it differs from the one in the 1609–1610 list.

24 Such were the cases of Gonçalo Monteiro, Gonçalo Garcia, Agostinho Gomes, João Mendes, Álvaro de Barros, João de Figueiredo, João Maciel, Gaspar Fernandes and António Mazarello or Mazaredo. ANTT, *TSO, CGSO*, liv. 369, fls. 34–34v.

25 This is expressed, for example, in some of the descriptions in the Reportorio, such as "to become a Gentile" (*se fazer gentio*).

26 ANTT, *TSO, CGSO*, liv. 369, fls. 27–27v.

27 BNRJ, Inquisição de Goa, 25,1,004 n.043, fls. 106–106v. In Portuguese: "Ceremonias, e ritos que de direito sam protestatiuas de Secta, como Çumbayas: que inuoluem o mayor acto de adoração dos Pagodes, que hâ entre os Gentios, sacrificios de fogo, sangue, ou qualquer outro, por que pretendem darlhes honra E Veneração como a Deos"; "outras muitas superstiçoes, e ritos, que não induzem tam efficax presumpção, nos quais se poderá sustentar o stillo antiguo, como ir as Cazas dos Pagodes, sem lhes fazer Çumbayas, assistir a suas festas, e bayles, e outros desta qualidade".

28 BNRJ, *Inquisição de Goa*, 25,1,004 n.154, fls. 357–357v. In Portuguese: "sacrifício, ou offerta, ou trato de idolatria, ou consulta E inuocação ao demonio"; "não são hereticais ou protestatiuas da seita gentilica, nem contem, suspe[i]ta de herezia, ou pacto com o demonio, mas somente h[ũa] superstição habitual nestes christãos da terra".

29 ANTT, *TSO, CGSO*, liv. 474; ANTT, *TSO, CGSO*, liv. 207, fols. 83–96 (transcribed in Ventura 2011, vol. 2, pp. 71–83).

30 In Portuguese: "ueneração & protestação da Religião dos dittos Pagodes & falsos Deoses"; "protestatiuos da Gentilidade."

31 In Portuguese: "por haereges & confiscados suas fazendas, & pode ser que alguns queimados". Figueira was less assertive in his statement about the condemnation of the *Cristãos da Terra* to capital punishment. In the end, his caution was justified. According to the data collected in the *Reportorio*, no individual belonging to this category of defendant was released to secular justice for behaviour associated with the practice of local cults or apostasy until 1623 (Thomaz 2018, p. 109). All the *Cristãos da Terra* who

received capital punishment were convicted of sodomy and not idolatry or any other similar offence. The exceptions, should we consider them as such, all pertained to individuals who left Goa with the purpose of avoiding the trial: António de Miranda (1589), sentenced to be relaxed in statue for absence, and D. Francisco de Noronha, sentenced to the same penalty because he fled during his trial in 1610. Also sentenced to be relaxed—although not technically *Cristãos da Terra*— were the neophytes Martim de Noronha (1574), a Jew who was baptised as an adult, for being guilty of practising Judaism, and Simão Ferreira (1585), also baptised as an adult from Islam, for relapsing in offences of "Moor" (*Mouro*) (BNP, Cód. 203, fols. 117, 198v, 269, 343, 372, 485v, 504v, 515v, 587v, 608v). The last two cases attest to a clear difference of approach regarding the treatment of neophytes from religions traditionally antagonistic to Catholicism, all the more so since general inquisitors such as D. Jorge de Almeida had recommended not condemning relapses until a dispensation from the pontiff had been obtained (ANTT, *TSO*, *CGSO*, liv. 311, fol. 91v).

[32] The fact that the Crown had enforced a general conversion of its Jewish and Muslim minorities left little space for visible signs of adherence to reproved religions in Portugal. Therefore, inquisitors in the tribunals of Lisbon, Évora and Coimbra mostly questioned those suspected of professing the "Law of Moses" or the "Sect of Muhammad" on less conspicuous actions such as fasting, prayers, and of course, circumcision. More noticeable gestural performances included praying against a wall using the "ataphalijs" (*tefillin*) (Judaism) or performing the "zala" or "sellaa" (*Salah*) (Islam) (*Collectorio* 1596, p. 5; Boronat y Barrachina 1901, t. 1, p. 226).

[33] BNRJ, *Inquisição de Goa*, 25,1,003 n.205, f. 421. In Portuguese: "as ditas insignias nam eram sinaes [pro]testatiuos dalgũa sei[ta], mas som*en*te politicos, p*er*a declarar e distinguir nobreza E sabed[o]ria".

[34] See the statements by Mateus Gomes Ferreira, notary of the Goa Inquisition, of 22 and 23 November 1632, and also the certificates produced by the secretary of the visit Fr. António de Andrade, SJ, of 3 and 4 January 1633. ANTT, *TSO*, *CGSO*, liv. 184, fols. 39v, 41, 94v, 97v.

[35] Certificates produced by the secretary of the visit Fr. António de Andrade, SJ, of 29, 30 and 31 December 1632, and 3 January 1633. ANTT, *TSO*, *CGSO*, liv. 184, fols. 89v, 91v, 94v, 96v.

[36] BNRJ, *Inquisição de Goa*, 25,1,004 n.020, fol. 48.

[37] Statement by António de Faria Machado of 10 November 1632, Visit to the Goa Inquisition by António de Vasconcelos. ANTT, *TSO*, *CGSO*, liv. 184, fol. 14.

[38] See the statement by Mateus Gomes Ferreira, notary of the Goa Inquisition, of 22 and 23 November 1632, and the certificates produced by the secretary of the visit Fr. António de Andrade, SJ, of 30 and 31 December 1632. ANTT, *TSO*, *CGSO*, liv. 184, fol. 39v, 41–41v, 92, 93v.

[39] BNRJ, *Inquisição de Goa*, 25,1,004 n.020, fol. 48. In Portuguese: "estillo que nessa jnquisicam ategora se guardou".

[40] In fact, it is still kept today among the archival holdings of the *Inquisição de Lisboa*. ANTT, *TSO*, *IL*, liv. 840, fols. 99–102. Published by Cunha (1995, pp. 295–301).

[41] ANTT, *TSO*, *CGSO*, liv. 100, fl. 100; Baião 1930, p. 271.

[42] In 1554, Inquisitor General Henrique elaborated the first instructions to serve as directives for an inquisitorial tribunal in Goa. A difficult-to-read marginal annotation on the document, mentioning the neophytes, not transcribed by Cunha when she published the instructions, seems to refer to the institutional framework practiced in the Spanish Inquisition with the "nuevamente convertidos de la secta de los moros" (newly converted from the sect of the Moors) or *moriscos*, who since 1530, at least, benefited from a papal grace that absolved *in utroque foro* all those who had apostatised (ANTT, *TSO*, *IL*, liv. 840, fol. 97; Boronat y Barrachina 1901, t. 1, pp. 135, 181).

[43] BNRJ, 025,01,001 n.177, fol. 397v.

[44] BNRJ, *Inquisição de Goa*, 25,1,004 n.043, f. 106–106v.

[45] BNRJ, *Inquisição de Goa*, 25,1,004 n.043, f. 106. In Portuguese: "os Jndios, que fazem ritos, e Cer[e]monias gentilicas, quando são protestatiuas, e Jndicatiuas violentamente da sua secta, deu*em* ser accusados pella tenção, por*que* a expe[r]ie*n*cia tem mostrado, que de ordinario as faz*em* com grande Crenca dos Jdolos, en cujo culto sam feitas: mas não seram accusados, quando os ritos, que fiserem, não forem desta q[u]alidade, e natureza".

[46] This may have resulted, in the short term, in a more hardened stance against *Cristãos da Terra*, for a few decades after this, a report by the ex-notary of the Goa Inquisition, Pedro Borges, to Alexander VII, mentions the first recorded condemnations of local Christians (up to 11 individuals) to the capital sentence by the tribunal during his eight years of service between late 1646, when Borges first arrived in Goa and early 1655, when he left for Rome by land (Sorge 1981, pp. 118–19; Thomaz 2018, p. 136).

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
