# Peer review of "From Idolatry to Gentilidade: Assessing Local Christians’ Religious Offences in the Goa Inquisition (17th Century)"

_religions, doi:10.3390/rel14121498_

Round 1

Reviewer 1 Report

Comments and Suggestions for Authors

The topic of this essay is so relevant that, perhaps, it would be useful to understand whether this category also stemmed from a better understanding of Brahaminian cults by the inquisitors, and whether the use of the term gentilidade is found in the theological and legal printed texts of the time, and from when. Perhaps it would also be useful to remark more clearly on the similarities between this category and the construction of the category of apostasy for New Christians and moriscos on Iberian soil.

Comments on the Quality of English Language

The English seems pretty good to me, but I suggest a final reading by an expert in this language.

Author Response

I have addressed this suggestion by adding the following sections to the article:

This is not to say that inquisitors at Goa were ill-informed on the specificity and functions of local rituals. Seeing as missionaries—who often served as deputies or inquisitors of the Goa Inquisition—not only wrote treatises on what they termed the “sects” of the Gentiles but were also learned on some of the great works of Hinduism such as the Bhagavadgītā, it is doubtful that the tribunal did not possess the necessary resources—human or documentary—to address issues pertaining to local rituals and devotions. In fact, when the Goa Inquisition was first abolished in 1774, its archive had no less than 7 manuscript books written in “Gentilic language” (lingoa gentilica) in the section of censored material, which points to some level of interaction with and knowledge of local literary religious expressions. Also, while the nature and intent of this list makes it difficult to ascertain which cults or practices were performed by the defendants, extant trials provide lengthier accounts of such rituals. A broad attempt at matching inquisitorial discourse and local rituals has never been pursued and falls outside the scope of this study. General letter of Brother Luís Fróis, SJ, to the Jesuits in Portugal, 8 December 1560, from Goa (Wicki 1956: 803); ANTT, TSO, CGSO, liv. 462, fols. 254-254v.

and

The fact that Portugal had enforced a general conversion of its Jewish and Muslim minorities left little space for visible signs of adherence to reproved religions. Therefore, inquisitors in Portuguese tribunals mostly questioned those suspected of professing the “Law of Moses” or the “Sect of Muhammad” on less conspicuous actions such as fasting, prayers, and of course, circumcision. More noticeable gestural performances included praying against a wall using the “ataphalijs” (tefillin) (Judaism) or performing the “zala” or “sellaa” (Salah) (Islam). Collectorio 1596: 5; Boronat y Barrachina 1901, t. 1: 226.

Reviewer 2 Report

Comments and Suggestions for Authors

The author could do a better job fleshing out the context of Indian religious history during this era. From the article, it is difficult to discern the actual practices (assuming Hindu praxis) that the inquisitors were encountering. The author elucidates the content of the inquisition sources, but the broader Indian religious context is lacking. In lines 49-51, the author notes that India presented unique challenges to the Portuguese inquisition, but they have not explained adequately what was unique about the sub-continent, as opposed to, say, Brazil. This is but one example of a general lack of adequate framing in the article. The author’s conclusions are potentially compelling, but this lack of framing leads to inadequate takeaway for the reader.

Citations throughout the article are erratic. Manuscripts appear to be cited in footnotes, while the literature is given in parenthetical citations (though sometimes mentioned in footnotes). Some citations read “author” or “blinded for review”: these must be clarified. It would be advisable to adopt a consistent citation practice throughout the article.

Comments on the Quality of English Language

The author writes with excessively wordy and sometimes vague prose. Overuse of passive verbal constructions and phrases like “with regard to” make reading this somewhat of a slog and at times obfuscate analytical points. Examples of wordy and unclear sentences can be seen in the following lines: 89-91, 99-102, 105-108, 123-127, 188-194, 482-486, 495-502, 511-515. A number of key points get buried inside of convoluted sentences and paragraphs. Some words and phrases become redundant very quickly—the word “context” is used five times in the second paragraph, for example. In the opinion of this reviewer, the entire article needs a grammatical and syntactical overhaul. The analysis of the sources is sound, but the prose does the author’s findings a serious disservice.

Author Response

Regarding the question of the unique challenges faced by the Goa Inquisition, the following section was added: While the Goa Inquisition was not the only tribunal to exert jurisdiction over colonial territories—the Inquisition of Lisbon’s district comprised American and African possessions—, the fact that its seat was established overseas to encompass an entirely colonial inquisitorial district meant that it was faced with a number of issues specific to the Estado da Índia. Missionary controversies, the participation of inquisitors in the Estado’s other forms of government, and vigilance over local Catholics’ involvement in native rituals were all problems that were unique to the Goa Inquisition. Because the inquisitors of the Lisbon tribunal were an ocean apart from Portugal’s African and American possessions, where local practices influenced the lives of Christianized populations, issues like missions, conversion, or participation in other state institutions never generated the kind of discussions it did in Goa. For this reason, the Goa Inquisition was the scene of challenges that were particular in the context of the Portuguese Holy Office, derived from the intertwining of colonial society and the tribunal.

Regarding the question of discerning local ritual practices, I addressed it by informing on the nature of the two practices explained by the Goa Inquisitors: The Portuguese usually employed the term “sumbaya” (from the Malay sembahyang) to refer to local ritual performances of reverential greetings or honor to both individuals and deities. As for the act of receiving a flower, it likely refers to the gifts first offered to a deity in a pūjā ceremony and then blessed by it (prasāda) and redistributed to the attendance. See Wilkinson, s. v. “sembahyang”, 405; Dalgado 1921, s. v. “Sumbaia, zumbaia”; Dalgado 1919, s. v. “fula”; Lidova 2020.

However, I should point out that a broad attempt at matching inquisitorial discourse and local rituals has never been pursued and was not the intent of this study, which focuses on European categories of religious offence. The inclusion of the table describing “transgressive behaviours” was meant to evince how the Holy Office translated behaviour into perceptible categories of religious offence for the purposes of conducting a trial. While the generic descriptions of such behaviours in the list does allow for some identifications, some leave us wondering still. To satisfactorily propose an identification of such rituals would necessitate a systematic comparison with extant inquisitorial trials and employ its own methodology and a lengthy discussion for each case (for instance, whether Bartolomeu Rangel’s behaviour is an expression of Tantric Hinduism or the result of it being absorbed into mainstream Brahmanism). To pursue it here would entail opening a section of its own, which I believe would divert the reader’s attention from the discussion I wish to conduct. I see it as an article/area of research on its own, which is why I opted for providing an explanation on those terms the inquisitors felt the need to clarify to the General Council.

Reviewer 3 Report

Comments and Suggestions for Authors

After reading the text submitted for evaluation, the following issues have been confirmed:

 Both the title, the summary and the keywords reflect the contents of the text. The work presents a correct scientific structure, which allows a simple and structured reading of it.

The work presents a very deep analysis of documentary sources related to judicial processes carried out by the Inquisition in Goa. The types of crimes and the change produced in the terminology are classified. The document presents a deep analysis of the documentation, statistical data and explanatory tables.

This is an important contribution to understanding the evolution and expansion of the Court of the Inquisition outside Europe and Latin America, and the consideration and classification of new crimes among the accused.

The work presents bibliographic references until the year 2021. It is recommended to include bibliographic references until the year 2023.

Author Response

I included references up to 2022. No new articles on the judicial practices of the Goa Inquisition concerning neophytes or cristãos da terra at large have been published – to my knowledge – in 2023.

Round 2

Reviewer 2 Report

Comments and Suggestions for Authors

The improvements to the prose and the clarifications on certain points have made the article much more compelling and have addressed the concerns I had about the previous version. I feel this is ready for publication.